# A Study on the Photoelectric Properties of Dual Ridge Terahertz Quantum Cascade Lasers at 3.1 THz

**DOI:** 10.3390/nano12152529

**Published:** 2022-07-23

**Authors:** Qi Yang, Jicheng Zhang, Xuemin Wang, Zhiqiang Zhan, Tao Jiang, Jia Li, Ruijiao Zou, Keyu Li, Fengwei Chen, Weidong Wu

**Affiliations:** Science and Technology on Plasma Physics Laboratory, Research Center of Laser Fusion, CAEP, Mianyang 621900, China; yangqi837@163.com (Q.Y.); zhangjccaep@126.com (J.Z.); wangxuemin@163.com (X.W.); zhanzhiqiangty@sina.com (Z.Z.); youlanhua@163.com (J.L.); zouruijiao@126.com (R.Z.); likeyu6969@163.com (K.L.); chenfengwei2007@126.com (F.C.); wuweidonging@163.com (W.W.)

**Keywords:** semiconductor lasers, terahertz, quantum cascade lasers, quantum well

## Abstract

High-power, incoherent THz array sources are widely desired in some applications due to their low energy, unique terahertz fingerprint, and image. In this work, a dual ridge terahertz quantum cascade laser lasing at 3.1 THz is presented, and the device’s performance is analyzed in detail. The maximum output power can reach 512 mW when the two ridges work simultaneously in continuous-wave mode, with a threshold current density of 281 A/cm^2^ at 15 K. While the peak power is approximately 704 mW in pulse-wave mode at 15 K, the lasing still could be observed approximately 7 mW at 125 K. The far-field pattern of the dual ridge THz QCL is detected by a THz camera; two light spots typically show a single-lobe Gaussian distribution. The experimental results provide a reference for realizing high-power THz quantum cascade lasers, and they will provide some guidance for the structural design of multiple ridges or laser arrays.

## 1. Introduction

Terahertz (THz) radiation spectrum locates at the transition region between microwave and optical frequency. Its frequency is usually defined in ranges from 0.1 to 10 THz (λ ≈ 3 mm~30 μm) [1]. Due to its low energy, unique terahertz fingerprint [2,3], and being transparent [4] to some materials, such as textiles, plastics, coatings, and biological tissues, it is a promising candidate for a variety of applications, such as remote sensing, spectroscopy, wireless communication, bio-medical, and so on [5,6,7,8,9,10].

Since the first THz QCL was demonstrated in 2002 [11], THz QCL has had a rapid development in the last decades. Compared with various available methods to generate THz radiation, terahertz quantum cascade lasers (QCLs) are one of the most promising sources due to their novel operation principle and their attainment of a performance that is compact, coherent, and efficient and shows a narrow emission linewidth [12,13,14]. Nowadays, the frequency of the THz QCLs ranges from 1.2 to 5.0 THz [15,16]. The highest output power in pulsed wave (PW) mode achieves more than 2.4 W [17], and the highest THz-QCL operating temperature reaches 250 K [18]. In order to promote its practical applications in some fields, a higher output power of THz QCL is desirable and much effort has been dedicated all along.

In general, by using broader area cavities, such as to increase the ridge width and the cavity length, the output power of the laser will have an obvious improvement, but scaling the device area to a certain value leads to a decrease in performance because of the Joule heat accumulation. On the other hand, THz QCLs could obtain a higher output power by adding periods of the active region. For example, using the wafer-bonding technique to stack two THz QCLs together, THz QCLs with a P_peak_ of up to 470 mW per facet at 5 K is achieved [19]. Another method is to grow a thicker active region, which achieves the highest output power so far [17]. While these methods increased the difficulty of wafer growth, which means an exact or a near exact active region thickness control during a longer growth time and a less more defect on the wafer’s surface. Hence, developing multiple ridges or arrays incorporated into the THz QCL sources may be an easier method to achieve a high output power. Up to now, it is still difficult to fabricate a single-lobe phased-locked THz QCL array, except for a surface-emitting structure [20]. Nevertheless, continuous wave high-power, incoherent THz array sources are still widely desired in applications, such as illuminating and real-time imaging systems [21].

In this study, a dual ridge THz QCL has been produced by using optical lithography and wet chemical etching technology, and the photoelectric properties of dual ridge THz QCL has been systematically studied. The results show that the device can obtain an output of several hundred milliwatts in both continuous wave (CW) mode and in pulse mode. The maximum power with a dual ridge could reach up to 512 mW with a threshold current density of 281 A/cm^2^ in CW mode at 15 K. The maximum pulse power is approximately 704 mW under the temperature, and the lasing still could be observed approximately 7 mW at 125 K.

## 2. Design and Fabrication

The active region structure design we used here is a hybrid bound to continuum transition and resonant phonon extraction, which is similar to that of Amaniti M.I. and Lianhe Li, et al. [22,23]. The structure was grown by solid source molecular beam epitaxy on a semi-insulating GaAs substrate based on a GaAs/Al_0.15_Ga_0.85_As material system, the device processing technology is similar to our previous work [24,25]. The structural parameters are shown in Table 1; the 200 periods of active region structures were sandwiched by 2 highly Si-doped GaAs layers.

In a typical device procedure, the laser wafer was monolithically processed into ridges by using optical lithography and wet chemical etching, the dimensions of each ridge was approximately 2.8 mm in length and 175 μm in width. A Ge/Au/Ni/Au (25/50/15/150 nm) layer was evaporated on top of the ridges and to form the bottom contact layer, which was 150 μm in width between the two ridges, and the bottom contact layer was 50 μm in distance apart from each ridge. A thermal annealing treatment (420 °C for 30 s) under nitrogen atmosphere provided a good ohmic contact. Then, a Ti/Au (10/200 nm) layer was deposited, covering both the bottom contact and the top of the dual ridges for Au wire bonding. The substrate was then thinned down to approximately 200 µm, and a Ti/Au layer was deposited on the backside for easy soldering. An Al_2_O_3_/Ti/Au/Ti/Al_2_O_3_ (200/10/100/10/200 nm) high-reflectivity(HR) coating was evaporated on the rear facet. The device was finally indium soldered onto the Cu heatsinks and bounded with Au wires.

## 3. Results and Discussion

A double-crystal X-ray diffraction measurement was used to examine the superlattice structure of the active region and its crystal growth quality along the crystallographic direction (hkl) = (004) of the laser wafer. The resolution of the X-ray diffractometer can reach 0.004° after calibration by a single crystal silicon standard sample and then the monochromated Cu Kα1 source scanned the wafer at 0.001 °/s under ω/2θ mode in the theta range from 31° to 35°.The HRXRD spectrum is shown in Figure 1, and a theoretical simulation curve for the design structure is also shown. The patterns revealed clear satellite peaks, which is associated with the superlattice structure of the active region. It could be seen that the satellite peaks marched well compared with the simulation result. The result demonstrates that actual active region thickness was in excellent agreement with the design, and the active region was under precision control during 200 periods by MBE growth techniques. It is noted from Figure 2 that the intensity of the high-angle satellite peaks in the experiment curve decays quickly, which may be due to interface roughness, diffuse scattering, detection noise, and so on in the active region.

The dual ridge THz QCLs were mounted on the cold finger of a liquid He flow cryostat with a polyethylene(PE) window for testing. The measured transmission of the window in THz range was approximately 71.7%. An f/2.0 OC-NI Winston cone was placed in front of the laser for enhancing collection efficiency. However, no correction is being presented in this paper for the transmission of the PE window and the collection efficiency.

The output power-current-voltage (P-I-V) characteristics of each ridge was performed at 15 K in CW mode and in PW mode. The output powers were collected by a room temperature detector (OPHIR 3A-P-THz). The results are shown in Figure 2. The dimensions of ridges we designed in this work are similar to our previous paper [24]. In the previous work, a batch of single ridge THz QCLs with different cavity lengths from 1.5 mm to 4.5 mm and the same width (175 μm) were fabricated. The result found that with the increase of cavity length, the laser output power will first increase and then decrease; the maximum output power appears near the 3 mm cavity length. So, we have designed the dual ridge THz QCL with 2.8 mm in length and 175 μm in width, as shown in the schematic diagram in Figure 2d, for the purpose of obtaining a high-power Terahertz laser source. Figure 2a,b presents the P-I-V characteristics of Ridge 1 and Ridge 2 in CW mode at 15 K; the peak optical power is approximately 289 mW and 272 mW; and the threshold current density of each ridge is approximately 286 A/cm^2^ and 276 A/cm^2^,respectively. Figure 2c shows the light power against the current characteristics of each ridge in pulse mode, the peak optical power is approximately 352 mW for Ridge 1 and 359 mW for Ridge 2, then the output power drops rapidly. Compared with the previous work [24], the output power of each ridge has a significant improvement. It may be due to the optimization of some process details, such as more accurate control of the epitaxial layer interface, higher thermal annealing temperature, etc.

The P-I-V performance of the two ridges working simultaneously has been investigated subsequently in CW mode, and results are shown in Figure 3a. A high collected peak optical power of 512 mW in CW mode is demonstrated with a threshold current density of 281 A/cm^2^ at 15 K. It can be found that the threshold current changes little compared with the single ridge, but the output power is only 91% of the sum of the maximum power of the two ridges because of the heat accumulation. The output power results show that when the dual ridges are working together, a higher output power could be obtained, which is almost the sum of the output power of each ridge. For spectrum measurement, the emissions were collected by a Fourier transform infrared (FTIR) spectrometer (SPS 300) and focused onto a room temperature deuterated triglycine sulfate (DTGS) detector. The spectra were collected in rapid scan mode with a resolution of 1 cm^−1^. After normalization of detection intensity, the result is shown in Figure 3b, from which one can see that the center frequency of the dual ridge lasers is approximately 3.1 THz. Besides the center frequency, one can note there are some small peaks around 3.1 THz, which is believed to be the multi-longitudinal mode excitation from the Fabry–Pérot cavity lasers.

Figure 4a shows peak output power against the current characteristics of the dual ridge THz quantum cascade laser at various heat-sink temperatures measured by conducting electric pulses with a pulse width of 20 μs and a frequency of 500 Hz to form a duty cycle of 1%. The peak output power is approximately 704 mW at 15 K, and lasing could be observed up to 125 K with an output power nearly 7 mW. To reduce the divergence angle, we employed an external convex lens with an effective focal length of 17 mm. In order to obtain the far-field pattern, a THz camera (MicroCAM^TM^ 3) is placed in front of the convex lens. Moreover, a 2.8 A current is input on the device by adjusting the position of the convex lens and the camera, two spots can be found on the screen, as shown in the inset of Figure 4b. Figure 4b shows the intensity distribution of the two light spots, from which one can find two separate quasi-Gaussian distribution spots, and in the center region of the spots there are two plateaus due to the oversaturation of the camera. What’s more, there are two rings around the light spots.

## 4. Conclusions

In this work, we have fabricated a dual ridge incoherent THz quantum cascade laser, A maximum power up to 512 mW is observed in CW mode at 15 K, and pulse output power at various heat-sink temperatures is also measured, The maximum power is approximately 704 mW, and the lasing still could be observed at 125 K. The experimental results shows that the output power changes a little when the two ridges operate in PW mode and in CW mode at the same time when the heat accumulation could be avoided. The light spots of the dual ridge THz QCL shows typically single-lobe Gaussian distribution by using an external convex lens. The experimental results show that the high-power laser is feasible, and it will provide some guidance for the structural design of multiple ridges or laser arrays.

## Figures and Tables

**Figure 1 nanomaterials-12-02529-f001:**
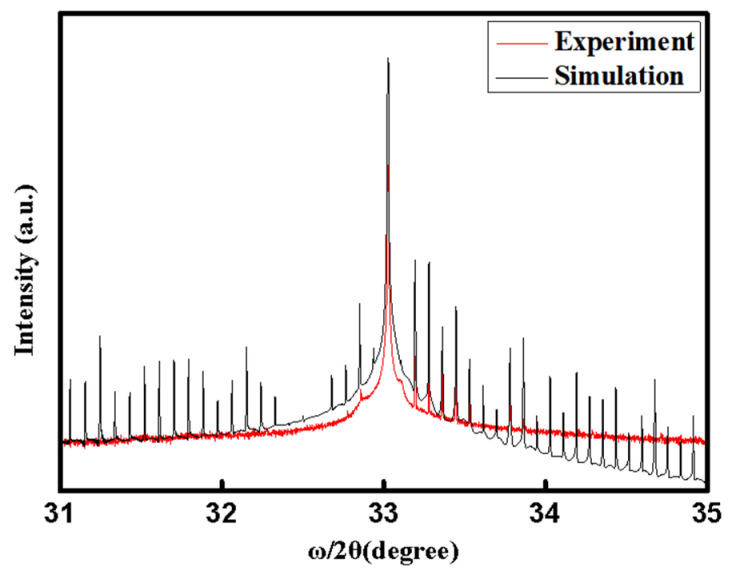
High resolution XRD spectrum and theoretical simulation curve of the wafer.

**Figure 2 nanomaterials-12-02529-f002:**
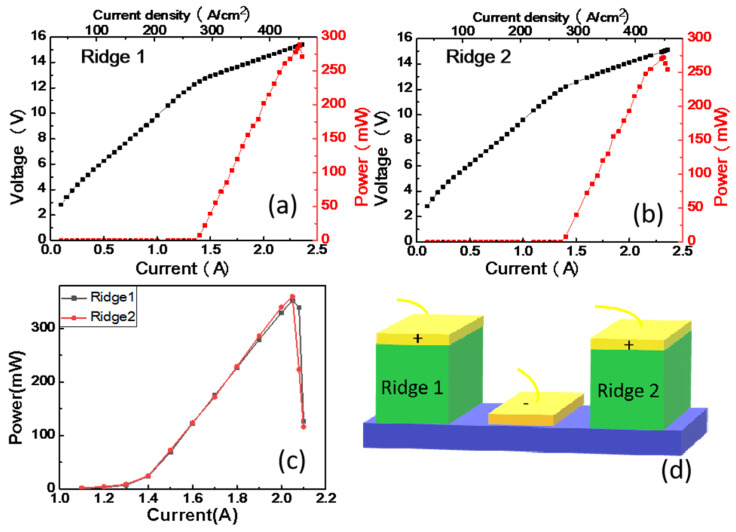
Typical P-I-V curves of each ridge in CW mode and in PW mode at 15 K. (**a**) P-I-V curve of Ridge 1 in CW mode; (**b**) P-I-V curve of Ridge 2 in CW mode; (**c**) P-I-V curves of two ridges in PW mode; (**d**) schematic diagram of the dual ridge THz QCL.

**Figure 3 nanomaterials-12-02529-f003:**
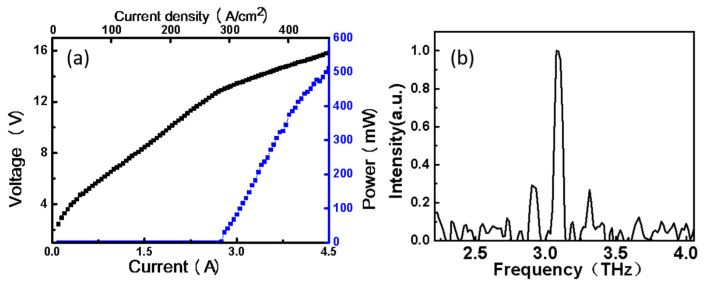
The P-I-V performance of dual ridges and typically multimode spectrum measured at 15 K. (**a**) Typical P-I-V curve of dual ridge device (2.8 mm × 175 μm); (**b**) The typical lasing spectrum at ~15 K.

**Figure 4 nanomaterials-12-02529-f004:**
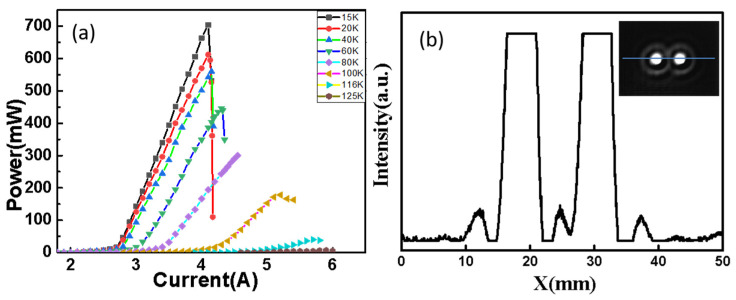
Pulsed P–I characteristics of dual ridge THz quantum cascade laser from 15 K to 125 K and light spot observed at 15 K. (**a**) Pulsed P–I characteristics of dual ridges; (**b**) Far-field patterns intensity distribution of dual ridges.

**Table 1 nanomaterials-12-02529-t001:** Structural parameters of GaAs/AlGaAs THz QCLs.

Constitution	Thickness	Doping	
GaAs	75 nm	5.0 × 10^18^ cm^−3^	contact layer
GaAs	17.2 nm	3.0 × 10^16^ cm^−3^	200 periods
Al_0.15_Ga_0.85_As	3.95 nm		
GaAs	8.8 nm		
Al_0.15_Ga_0.85_As	3.6 nm		
GaAs	10.75 nm		
Al_0.15_Ga_0.85_As	1.7 nm		
GaAs	10.3 nm		
Al_0.15_Ga_0.85_As	5.2 nm		
GaAs	700 nm	2.5 × 10^18^ cm^–3^	buffer
S.I. GaAs wafer			

## Data Availability

Not applicable.

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
