# Peer review of "A Study on the Photoelectric Properties of Dual Ridge Terahertz Quantum Cascade Lasers at 3.1 THz"

_nanomaterials, 2022, doi:10.3390/nano12152529_

Round 1
Reviewer 1 Report
This paper reports the improvement of a THz quantum cascade laser mainly in output power. Although the manuscript is well written, I feel some information is missing.
Since the improvement compared with author's previous work is mainly the efforts in the geometrical design of the device, the authors should add a figure for schematically explanation. The shape and the dimension of the device are important information.
The authors inspected the device by high-resolution XRD. However, the measurement condition including the type of the X-ray source is not explained. It seems that the authors used not synchrotron source but conventional one, but the kinds of the anode target is not mentioned. The authors explain the scanning speed as the performance, but it is not angular resolution itself.
In Fig. 1 the both rocking curves should be normalized with respect to the vertical axis for comparison between the simulation and the experiment. Indeed, several peaks in the rocking curves are consistent with each other, but many peaks in the simulation are not found in the experimental curve. The authors should briefly explain the result.
The spectrum of output THz-wave is inserted to Fig.3. I strongly recommend that the spectrum should be plotted in individual figure for detailed explanation. On the other hand, the photograph of a power meter in Fig.3 is not essential.
Author Response
Open Review 1
Comments and Suggestions for Authors
This paper reports the improvement of a THz quantum cascade laser mainly in output power. Although the manuscript is well written, I feel some information is missing.
Since the improvement compared with author's previous work is mainly the efforts in the geometrical design of the device, the authors should add a figure for schematically explanation. The shape and the dimension of the device are important information.
Answer: Thank you for your suggestion and we have already added a figure to explain the device structure as shown inset Fig.2c. (manuscripts in line 118)
The authors inspected the device by high-resolution XRD. However, the measurement condition including the type of the X-ray source is not explained. It seems that the authors used not synchrotron source but conventional one, but the kinds of the anode target is not mentioned. The authors explain the scanning speed as the performance, but it is not angular resolution itself.
Answer: Thank you for your question, the monochromated Cu Kα1 source is used to examine superlattice structure of the active region and its crystal growth quality, and the resolution of X-ray diffractometer can reach 0.004° after calibration by single crystal silicon standard sample. (manuscripts in line 83-88)
In Fig. 1 the both rocking curves should be normalized with respect to the vertical axis for comparison between the simulation and the experiment. Indeed, several peaks in the rocking curves are consistent with each other, but many peaks in the simulation are not found in the experimental curve. The authors should briefly explain the result.
Answer: Thank you for your question, the both rocking curves has been normalized and the XRD pattern was reprocessed as shown in Fig. 1. Compared with simulation curve, the decreasing intensity of the high-angle satellite peak in the experiment curve might be derived from interface roughness, diffuse scattering, detection noise and so on. (manuscripts in line 94-97)
The spectrum of output THz-wave is inserted to Fig.3. I strongly recommend that the spectrum should be plotted in individual figure for detailed explanation. On the other hand, the photograph of a power meter in Fig.3 is not essential.
Answer: Thank you for your suggestion, the spectrum has been plotted in a individual figure and detailed explanation was given. And the photograph of the power meter in Fig.3 has been deleted.(manuscripts in line 137-150)
Reviewer 2 Report
The authors created and conducted a study of a compact source of terahertz radiation, useful for the development of modern terahertz technologies. Significant power levels have been obtained and ways of its further increase have been indicated.
The article, in my opinion, needs some improvement.
First, it is very carelessly written and framed. The authors globally ignore spaces between numbers and units, confuse upper and lower case letters, etc. (see yellow marks in the file).
Secondly, the quality of drawings also leaves much to be desired (see ibid.).
Thirdly, keywords "dual ridges" and "high power" by themselves do not characterize anything.
As for the physical content, I have suggestions for improvement in the following three points:
1) The place of this type of terahertz radiation source among other terahertz sources should be more clearly indicated. It is clear that such a source has one undeniable advantage - its compactness with relatively high power. It seems that in other parameters (monochromaticity, noise, stability) it loses a lot. However, for example, for terahertz imaging systems, such property as incoherence will be not only not harmful, but even useful due to the lack of a spectral pattern that occurs with coherent sources.
2) The authors should describe in more detail the spectral properties of the radiation (see notes to Fig. 3 in file).
3) The authors should refine and explain the image of radiation beams (see notes to Fig. 4 in file).

Author Response
Open Review 2
Comments and Suggestions for Authors
The authors created and conducted a study of a compact source of terahertz radiation, useful for the development of modern terahertz technologies. Significant power levels have been obtained and ways of its further increase have been indicated.
The article, in my opinion, needs some improvement.
First, it is very carelessly written and framed. The authors globally ignore spaces between numbers and units, confuse upper and lower case letters, etc. (see yellow marks in the file).
Answer: Thank you for your carefully check, the yellow marks in the file have already been revised carefully.
Secondly, the quality of drawings also leaves much to be desired (see ibid.).
Answer: Thank you for your suggestion, we have already replaced all those figures.
Thirdly, keywords "dual ridges" and "high power" by themselves do not characterize anything.
Answer: Thank you for your question, we have modified the keywords of the manuscript. (manuscripts in line 18)
As for the physical content, I have suggestions for improvement in the following three points:
- The place of this type of terahertz radiation source among other terahertz sources should be more clearly indicated. It is clear that such a source has one undeniable advantage - its compactness with relatively high power.It seems that in other parameters (monochromaticity, noise, stability) it loses a lot. However, for example, for terahertz imaging systems, such property as incoherence will be not only not harmful, but even useful due to the lack of a spectral pattern that occurs with coherent sources.
Answer: Thank you for your suggestion and it has been modified in the manuscripts.
- The authors should describe in more detail the spectral properties of the radiation (see notes to Fig. 3 in file).
Answer: Thank you for your question, the spectrum has been plotted in a individual figure and a detailed explanation was made. (manuscripts in line 137-150)
- The authors should refine and explain the image of radiation beams (see notes to Fig. 4 in file).
Answer: Thank you for your question, Detail information about how to obtain the far-field pattern was discussed in our revised manuscript, and the intensity distribution on center position of light spots is obtained.(manuscripts in line 155-164)